# Sex-specific distribution and classification of *Wolbachia* infections and mitochondrial DNA haplogroups in *Aedes albopictus* from the Indo-Pacific

**Qiong Yang** [1]☯*, **Jessica Chung** [1,2]☯, **Katie L. Robinson** [1], **Thomas L. Schmidt** [1], **Perran A. Ross** [1], **Jiaxin Liang** [1], **Ary A. Hoffmann** [1]*

**1** Pest and Environmental Adaptation Research Group, Bio21 Institute and the School of BioSciences, The University of Melbourne, Parkville, Australia, **2** Melbourne Bioinformatics, The University of Melbourne, Parkville, Australia

☯ These authors contributed equally to this work.
* qiongy@unimelb.edu.au (QY); ary@unimelb.edu.au (AAH)

**Data Availability Statement:** All relevant data are within the manuscript and its Supporting Information files.

## Abstract

The arbovirus vector *Aedes albopictus* (Asian tiger mosquito) is common throughout the Indo-Pacific region, where most global dengue transmission occurs. We analysed population genomic data and tested for cryptic species in 160 *Ae. albopictus* sampled from 16 locations across this region. We found no evidence of cryptic *Ae. albopictus* but found multiple intraspecific COI haplotypes partitioned into groups representing three Asian lineages: East Asia, Southeast Asia and Indonesia. Papua New Guinea (PNG), Vanuatu and Christmas Island shared recent coancestry, and Indonesia and Timor-Leste were likely invaded from East Asia. We used a machine learning trained on morphologically sexed samples to classify sexes using multiple genetic features and then characterized the *w*AlbA and *w*AlbB *Wolbachia* infections in 664 other samples. The *w*AlbA and *w*AlbB infections as detected by qPCR showed markedly different patterns in the sexes. For females, most populations had a very high double infection incidence, with 67% being the lowest value (from Timor-Leste). For males, the incidence of double infections ranged from 100% (PNG) to 0% (Vanuatu). Only 6 females were infected solely by the *w*AlbA infection, while rare uninfected mosquitoes were found in both sexes. The *w*AlbA and *w*AlbB densities varied significantly among populations. For mosquitoes from Torres Strait and Vietnam, the *w*AlbB density was similar in single-infected and superinfected (*w*AlbA and *w*AlbB) mosquitoes. There was a positive association between *w*AlbA and *w*AlbB infection densities in superinfected *Ae. albopictus*. Our findings provide no evidence of cryptic species of *Ae. albopictus* in the region and suggest site-specific factors influencing the incidence of *Wolbachia* infections and their densities. We also demonstrate the usefulness of ddRAD tag depths as sex-specific mosquito markers. The results provide baseline data for the exploitation of *Wolbachia*-induced cytoplasmic incompatibility (CI) in dengue control.

**Funding:** AAH was supported by program and Fellowship grants from the National Health and Medical Research Council (NHMRC), nos. 1132412 and 1118640 (https://www.nhmrc.gov.au). The funders had no role in study design, data collection and analysis, decision to publish, or preparation of the manuscript.

## Author summary

The mosquito *Aedes albopictus* transmits dengue and other arboviruses. This study investigates the genetics of these mosquitoes and their endosymbiont *Wolbachia* in the Indo-Pacific region, where 70% of global dengue transmission occurs. The analysis of mitochondrial DNA sequences showed no evidence of cryptic *Ae. albopictus* but suggested three Asian lineages: East Asia, Southeast Asia and Indonesia. Papua New Guinea, Vanuatu and Christmas Island shared recent coancestry, and Indonesia and Timor-Leste were likely invaded from East Asia. We used bioinformatics to classify sexes and then characterized the *w*AlbA and *w*AlbB *Wolbachia* infections via both bioinformatics and quantitative PCR. We found markedly different patterns of *w*AlbA and *w*AlbB infections in the sexes. The *w*AlbA and *w*AlbB densities varied significantly among populations, suggesting site-specific factors influencing the incidence of *Wolbachia* infections and their densities. We also demonstrate the usefulness of next generation sequencing data in developing molecular markers that can be repeatedly reanalysed to investigate new issues as these arise. These results provide baseline data for the exploitation of *Wolbachia*-induced cytoplasmic incompatibility in dengue control.

## Introduction

The mosquito *Aedes albopictus* is an important disease vector, capable of transmitting dengue, chikingunya and other arboviruses. As such, it has been targeted for control in many countries around the world, including areas where it has invaded relatively recently from its origin in east Asia such as North America and Europe [1]. *Aedes albopictus* is common throughout the Indo-Pacific region, where 70% of global dengue transmission occurs [2]. Control is challenging due to a variety of factors including the evolution of pesticide resistance [3,4] and the wide range of hosts and habitats used by *Ae. albopictus* [5], which has led to renewed interest in developing alternative methods of control and disease suppression. These include using the endosymbiotic bacterium *Wolbachia*, where males carrying a novel *Wolbachia* infection are released to sterilise females [6–8], or to spread through populations to reduce arboviral transmission by the mosquitoes [9,10].

When applying *Wolbachia* technology, it is important to understand both the taxonomic identity of the mosquito target as well as the status of natural *Wolbachia* infections in the mosquito target. Both factors are important in the case of *Ae. albopictus*, as a cryptic *Ae. albopictus* subspecies has been described from China [11] and Vietnam [12] and because *Ae. albopictus* in the field naturally harbor the *Wolbachia* strains *w*AlbA or *w*AlbB (and are often superinfected with both strains) [13,14]. Patterns of population genetic structure can indicate areas where mosquito genetic backgrounds are likely to be similar or different to those of target populations [15]–an important consideration for widespread species such as *Ae. albopictus* that can be locally adapted to different conditions [16].

Both suppression and invasion of populations by novel *Wolbachia* depend on interactions between the novel strain and extant *Wolbachia* strains as well as their impacts on host fitness and transmission efficiency [17]. In *Ae. albopictus*, *w*AlbA and *w*AlbB have been reported as increasing host fecundity [18]. Both *w*AlbA and *w*AlbB infections also induce cytoplasmic incompatibility (CI), the phenomenon where males carrying a particular *Wolbachia* strain are incompatible with females lacking that strain, resulting in the production of unviable embryos, which is a common phenotype in insects [19]. Females infected with only *w*AlbA or *w*AlbB strain show CI when crossed with superinfected males resulting in unidirectional CI [13].

However, this CI phenotype may be rarely expressed in nature if there is a high frequency of individuals carrying both *w*AlbA and *w*AlbB and a low level of polymorphism within these infections [20]. Other factors influencing the incidence of these infections in natural populations include maternal transmission efficiency which can be imperfect for *Wolbachia* [19] and may be somewhat lower for the *w*AlbA strain than for *w*AlbB (estimates of 97.5 versus 99.6%, respectively for one region) [14]. Both fitness effects and transmission of *Wolbachia* may relate partly to *Wolbachia* density; high densities of endosymbionts are more likely to decrease host fitness but maintain a high level of maternal transmission and CI [17]. Moreover, variability in density is important because crosses between individuals with the same strain but different densities can produce CI [21].

In this paper, we build on existing work to provide an overview of *w*AlbA and *w*AlbB infections among *Ae. albopictus* in the Indo-Pacific region. We aim to characterise the incidence of these infections from field collections across the region. We also test for the cryptic species status of the collections given that *Wolbachia* may be absent from one of the cryptic subspecies [12]. We focus on any sex-related differences in infection frequencies given that the incidence of *w*AlbA in particular may differ between males and females [22,23]. To test associations between *Wolbachia* and sex in samples that were only available to us without sexing (e.g. larval samples, DNA samples), we develop a method based on previously collected ddRAD sequencing data to sex the mosquitoes given that the sexes of *Aedes* mosquitoes can be differentiated through sequencing depth at multiple pseudosex regions [24]. Finally, we consider variation in the field density of the *Wolbachia* infections to test whether there are possible interactions between the infections given that the presence of one endosymbiont in an invertebrate host can influence the density of another endosymbiont [25]. We discuss our results with respect to future *Wolbachia*-based strategies against *Ae. albopictus*.

## Material and methods

### Sample collection

*Aedes albopictus* were sampled from 17 locations from Mauritius to Fiji to Japan (Fig 1 and Table 1). We considered mosquitoes collected within the same country to be from the same population. Genome sequencing data obtained from a total of 664 *Ae. albopictus* individuals among these populations have been used for population genetic analysis in our previous study [15], where genotypes were filtered using a bioinformatics pipeline. Here, we revisited those ddRAD sequencing data and used sequencing depths and SNPs to predict sex and classify *Wolbachia* infection status by sex (see below).

### Mitochondrial COI as a DNA barcode

The universal barcode region of the CO1 gene (658 bp) of individual *Ae. albopictus* was amplified using the common primers (LCO1490 5' GGTCAACAAATCATAAAGATATTGG 3' and HCO2198 5' TAAACTTCAGGGTGACCAAAAAATCA 3') [26]. Amplifications were performed in a Thermal Cycler (Eppendorf, Germany) with an adjusted annealing temperature of 55˚C. PCR amplicons from individuals were sequenced in both forward and reverse directions using Sanger Sequencing (Macrogen, Inc., Geumcheongu, Seoul, South Korea). The trimmed 623 bp sequence was analysed with Geneious 9.18 software to investigate SNP variation among samples.

The CO1 gene sequencing data obtained from 160 mosquitoes were analysed with the Molecular Evolutionary Genetics Analysis (MEGA) program version 7.0. A phylogenetic tree was constructed with a neighbour-joining model applied to a genetic distance matrix with the Kimura-2 parameter model implemented with 1000 bootstrap replications in MEGA.

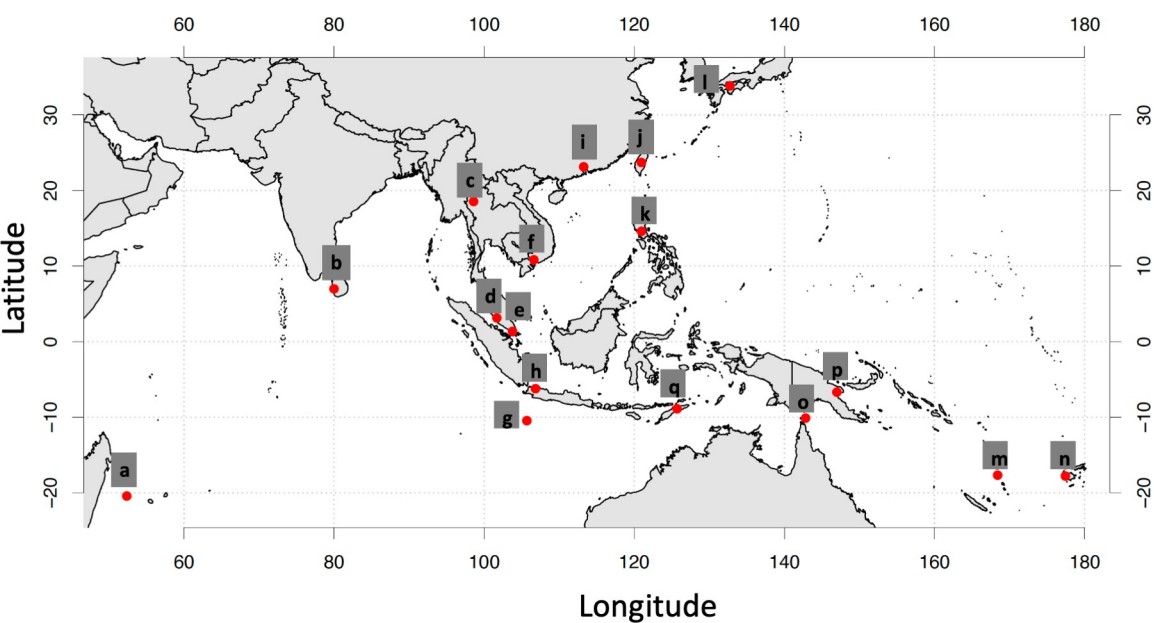

**Fig 1. Approximate locations of the 17 *Aedes albopictus* populations.** The map was made with R Package mapdata. Each letter corresponds to a collection from a different country. Note that in some cases multiple samples from nearby locations were combined (Table 1).

## ML sex classification with ddRAD-seq

To build a machine learning (ML) classifier, we used a semi-supervised learning approach with 91 individuals from Torres Strait that had both sexing based on morphology and ddRAD sequencing data from our previous study, and 134 unsexed samples with ddRAD sequencing data only. For each sample BWA-MEM v0.7.17 [27] was used to map sequencing reads to the

**Table 1. Details of *Aedes albopictus* sampled from 17 populations.** See Fig 1 for map ID locations.

| Map ID | Country | Year(s) collected | Locations combined | Life stage |
|---|---|---|---|---|
| a | Mauritius | 2017 | | adult |
| b | Sri Lanka | 2017 | | larva |
| c | Thailand | 2016 | | adult |
| d | Malaysia | 2015/2017 | Pahang, Johor, Selangor and Perlis | adult |
| e | Singapore | 2015 | | adult |
| f | Vietnam | 2014 | | adult |
| g | Christmas Island | 2018 | | larva |
| h | Indonesia | 2016/2017 | Jakarta, Bandung and Bali | Jakarta and Bandung: adult, Bali: larva |
| i | China | 2017 | | adult |
| j | Taiwan | 2016 | | most likely adults |
| k | Philippines | 2016 | | most likely adult |
| l | Japan | 2015 | | adult |
| m | Vanuatu | 2018 | | most likely adults |
| n | Fiji | 2018 | | most likely adult |
| o | Torres Strait | 2018 | multiple islands | adult |
| p | Papua New Guinea (PNG) | 2019 | Port Moresby and Madang | adult |
| q | Timor-Leste | 2019 | | adult |

*Ae. albopictus* reference genome (GenBank accession no. GCA_006496715.1) and determined the relative sequencing depths of 26,782 100kbp sized regions. For each of these 100kbp regions, a linear model was fit to identify if sex had a significant effect on depth, using sex and sequencing batch as predictor variables. We identified 85 regions with significant sex effects after multiple hypothesis testing correction. A PCA plot using the sequencing depths of significant regions showed two distinct clusters (S1 Fig), however, we observed 15 individuals not clustering with their expected group suggesting some samples may have been assigned the incorrect sex. These samples were removed from the training set. Using the e1071 R library [28], a preliminary SVM model was trained using the significant regions as features and the 135 unsexed samples were classified using this model (S2 Fig).

For the purposes of creating a stand-alone program for classifying new ddRAD samples, we used samples with classification probability > 70% from the preliminary model and extracted candidate ddRAD tags within the identified 100kbp regions as candidate features. From these, we used linear models to identify 42 ddRAD tag features that had their sequencing depths associated with sex. In addition, 109 negative control ddRAD tags not associated with sex were included for sequence depth normalisation purposes. A SVM model was trained using the e1071 library and the classifier has been made available on GitHub (https://github.com/pearg/albo_spm).

To validate the model, we obtained an additional 127 sexed individuals from different populations (Indonesia, Japan, Vietnam, and Torres Strait). 124 of these had sufficient sequencing depth to predict sex, and we compared the sex predicted from our model to the sex obtained from morphology. A confusion matrix (predicted versus actual cases) was used to test the quality of the classification model with a Matthews correlation coefficient computed to quantify the agreement.

### ML *Wolbachia* infection classification with ddRAD-seq

Due to insufficient observations of uninfected samples and samples infected with *w*AlbA only, we limited our classifier to single *w*AlbB infected samples and superinfected (*w*AlbA and *w*AlbB) samples. To build the *Wolbachia* strain classifier, we first obtained SNP sites between the *w*AlbA strain and the *w*AlbB strain by comparing the *Wolbachia w*AlbA FL2016 strain contig-level assembly (GenBank accession no. GCA_002379155.2) to the *w*AlbB complete assembly (GenBank accession no. GCA_004171285.1) by simulating 100bp reads from the *w*AlbA reference and mapping them to the *w*AlbB reference. Using only the uniquely mapping reads, we used samtools mpileup [29] to obtain sites with single nucleotide differences between the two references, resulting in 9,697 sites.

*Wolbachia*-infected *Ae. albopictus* ddRAD-seq samples were aligned using Bowtie2 v2.3.4.3 [30] to the *w*AlbB reference assembly and variant calling was performed with FreeBayes v1.3.5 [31]. We then selected SNPs that were previously identified in the *w*AlbA/*w*AlbB reference assembly comparison, resulting in 649 SNP sites. Samples with fewer than 10 allelic observations were removed, leaving 478 *w*AlbB infected (n = 63) or superinfected (n = 415) samples. The data was split into a training and test set of 80% and 20% respectively.

For each sample, a score was created using the number of observed *w*AlbA alleles divided by the total number of alleles observed, then Laplace smoothed with a pseudocount of 1. A univariate logistic regression model was built using the natural log of the score. The test dataset was used to assess the performance of the model. As with the sexing prediction, a confusion matrix (predicted versus actual cases) was used to test the performance of the classification model.

### *Wolbachia* detection via qPCR assay

For real-time PCR detection, we used a LightCycler 480 High Resolution Melting Master (HRMM) kit (Roche; Cat. No. 04909631001, Roche Diagnostics Australia Pty. Ltd., Castle Hill

New South Wales, Australia) and IMMOLASE DNA polymerase (5 U/μl) (Bioline; Cat. No. BIO-21047) as described by Lee et al. (2012) [32] which we have used effectively in a variety of contexts [21,33]. The PCR conditions for DNA amplification began with a 10-minute pre-incubation at 95˚C (Ramp Rate = 4.8˚C/s), followed by 40 cycles of 95˚C for 5 seconds (Ramp Rate = 4.8˚C/s), 53˚C for 15 seconds (Ramp Rate = 2.5˚C/s), and 72˚C for 30 seconds (Ramp Rate = 4.8˚C/s).

Primers specific for the gene encoding the *Ae. albopictus* internal transcribed spacer (ITS2) (forward primer: 5'-GCATGGCCAACCTCTAGC-3', reverse primer: 5'-CCGCCACTTAGC TATGTCAA-3') was used to confirm that individual mosquitoes were correctly identified *as Ae. albopictus* and to normalise for differences in mosquito size. Primers specific to either *w*AlbA (forward primer: 5'-GTAGTATTTACCCCAGCAG-3', reverse primer: 5'- CACCAG CTTTTACTTGACC-3') or *w*AlbB (forward primer: 5'- CCTTACCTCCTGCACAACAA-3', reverse primer: 5'- GGATTGTCCAGTGGCCTTA-3') were used to infer the presence or absence of *w*AlbA and *w*AlbB [34]. Crossing point (Cp) values of three consistent replicate runs were averaged to produce the results. Differences in Cp values between the *Ae. albopictus* marker and the *w*AlbA and *w*AlbB markers were transformed by $2^n$ to produce relative *Wolbachia* density measures.

## Statistical analyses

We compared the proportion of superinfected individuals among the sexes and populations using a generalized linear model with a binomial distribution. We also directly compared the distribution of superinfected and *w*AlbB singly infected individuals across sexes in some populations with relatively larger sample sizes, treating these as contingency tables. All these analyses were run in IBM Statistics SPSS version 26.

For the *Wolbachia* density data, we examined variation in density across populations and sexes following logarithmic transformation of the data for normality, focussing on those individuals carrying both infections. We also examined associations between *w*AlbA and *w*AlbB infection density at the individual level within each population by computing Pearson's correlations on logarithmically transformed densities and treating the sexes separately given the density differences observed between the sexes (see below). Linear regressions were also computed to see if the density of *w*AlbA could predict that of *w*AlbB. We only considered correlations and regressions for populations where data from at least 10 individuals were available. Finally, we tested if there was a difference between *w*AlbB densities in singly and superinfected females and males; we focussed on the Torres Strait and Vietnam populations, where samples falling into both categories were available, and included sex and infection type as factors for Torres Strait and only infection type in Vietnam since only females had sufficient numbers of each infection type.

## Results

### Variation in *Ae. albopictus*

PCR amplification and sequencing of the mtDNA CO1 gene resulted in a 623 bp fragment for each individual *Ae. albopictus* with no insertions or deletions. A total of 23 haplotypes were identified from 160 individuals collected in the Indo-Pacific region. Haplotype diversity varied for each region with Indonesia (31 individuals, 8 haplotypes), Vietnam (16 individuals, 6 haplotypes) and Thailand (8 individuals, 5 haplotypes) having substantially higher haplotype diversities than Taiwan (14 individuals, 3 haplotypes), Sri Lanka (14 individuals, 2 haplotypes), Malaysia (12 individuals, 2 haplotypes), PNG (11 individuals, 2 haplotypes), Singapore (8 individuals, 2 haplotypes) and Vanuatu (19 individuals, 1 haplotype).

Three predominant haplotypes were identified in *Ae. albopictus* populations: H1 (17.5%) detected in Vanuatu, PNG and Christmas Island, suggesting recent co-ancestry; H2 (15.0%) detected in Indonesia, Torres Strait, Timor-Leste and Philippines; and H11 (18.8%) detected in a wide range of populations, including Malaysia, Singapore, Thailand, Indonesia, Fiji, Philippines, Vietnam and PNG. Other haplotypes were either unique to a specific population or had a limited geographical coverage.

To determine the relationships among samples, we constructed a median-joining network using haplotypes based on sequence variation. Haplotypes were connected when the probability of parsimony was at least 95%. The COI haplotype network (Fig 2) suggested some mitochondrial genetic structure between regions. The spatial distribution of ancestral lineages among *Ae. albopictus* can be interpreted with reference to the native and invasive ranges of this species. A network analysis of *Ae. albopictus* partitioned haplotypes into three haplogroups representing three native range lineages. The northernmost of these (red dotted circle) showed a common heritage among East Asian populations. This lineage was also dominant in Mauritius. The second native range lineage (blue dotted circle) had a common heritage among Southeast Asian populations (excluding Indonesia), and was also found in Christmas Island, Fiji, PNG and Vanuatu. Differentiation was low between these lineages. The Indonesian *Ae. albopictus* lineage (green dotted circle) was distinct from the East and Southeast Asian native range lineages and was also found in the Torres Strait Islands and Timor-Leste. The Philippine haplotypes were split into two groups, Southeast Asia and Indonesia. This likely reflects the Philippine population as having ancestry from both groups. A further split was observed in the phylogenetic tree based on CO1 sequence variation from the 23 haplotypes in this study, the 2 *Ae. albopictus* cryptic haplotypes (KY765450.1 and KY765459) [11], as well as the outgroup *Ae. scutellaris* (KP843372, Fig 3). The maximum divergence observed in the 23 haplotypes was 1.6% (S1 Table), significantly lower than interspecific divergence (12.8%), indicating that cryptic *Ae. albopictus* are not present across the samples of the Indo-Pacific region tested here.

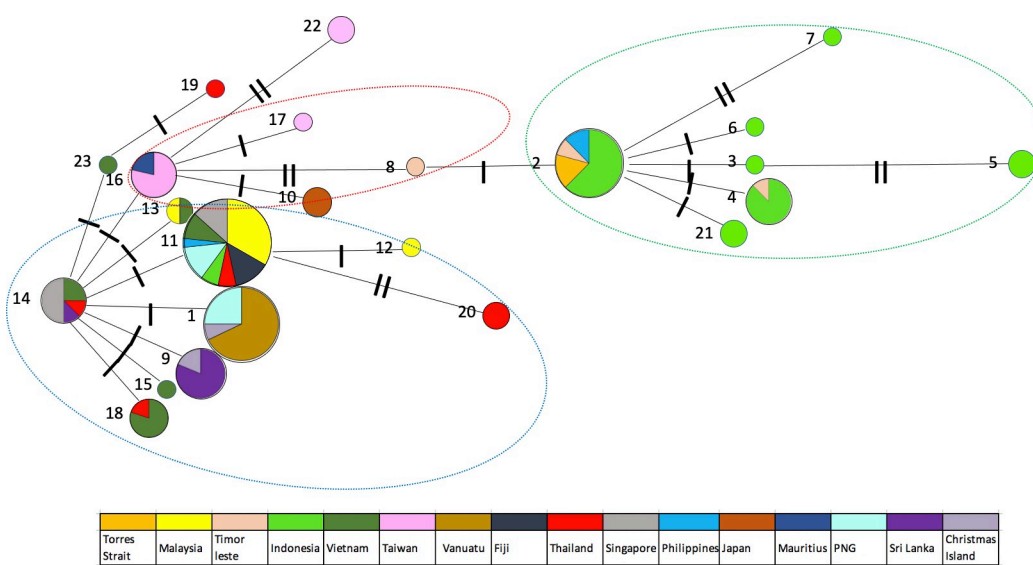

**Fig 2. Haplotype network for COI.** Each coloured node represents an observed haplotype with circle size indicating the number of individuals with each numbered haplotype and the slash along the connecting lines indicating the number of nucleotide differences.

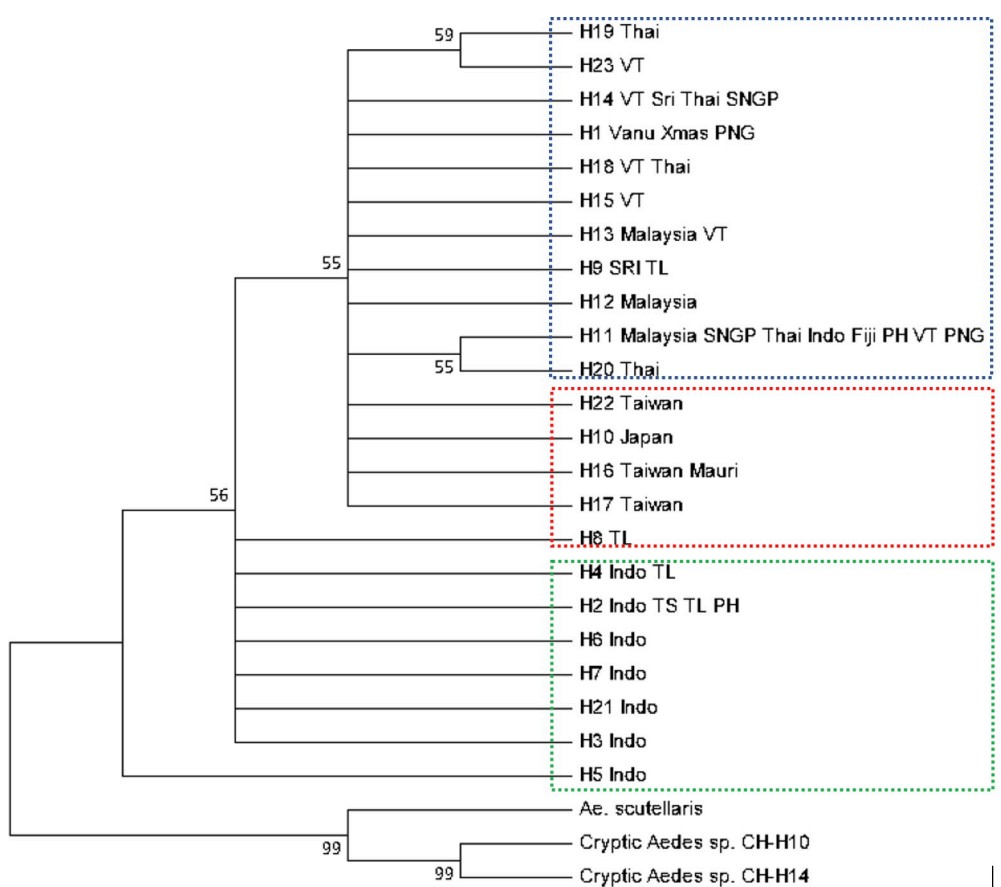

**Fig 3. Phylogenetic analysis based on CO1 haplotype variation.** Neighbor-joining trees constructed via Kimura-2 parameter model using MEGA. Numbers at branches represent bootstrap values of 1000 replicates (values > 50 are shown). Sequences from different outgroup species of the genus *Aedes* were selected from GenBank (*Ae. albopictus* cryptic haplotypes KY765450.1 and KY765459 and *Ae. scutellaris* KP843372). Abbreviations: Thai, Thailand; VT, Vietnam; Sri, Sri Lanka; SNGP, Singapore; Vanu, Vanuatu; Xmas, Christmas Island; PNG, Papua New Guinea; TL, Timor-Leste; TS, Torres Strait; Indo, Indonesia; PH, Philippines.

## Sex determination

As shown in Table 2, the classification performance on the 124 samples in the test set yielded a model accuracy of 97.6% with three incorrect classifications. The model had a Matthews correlation coefficient (MCC) of 0.940 and it is a relatively reliable method for sex classification.

We then used our model to classify a further 510 samples without morphological sexing (Table 3, excluding Torres Strait samples).

## *Wolbachia* infection classification

As shown in Table 4, the classification performance of the 95 samples in the test set yielded a model accuracy of 87.3%. However, the dataset was imbalanced with most samples being superinfected, thus only yielding an MCC of 0.634. The majority of the discordant samples were predicted to be superinfected but were only scored as singly infected with *w*AlbB in the qPCR assay, producing the low MCC value. We suspect that this relates to the limit of detection of *w*AlbA in the qPCR assay rather than inaccurate classification of the samples (see Discussion).

## Sex-specific distribution and classification of *Wolbachia* infection

A total of 664 *Ae. albopictus* individuals among 15 populations were processed and screened for the presence of *Wolbachia* using qPCR. Overall, *w*AlbA and *w*AlbB infections showed markedly different patterns with regard to sex (Table 3). The incidence of superinfected individuals was almost always higher in females than in males. For females, most of the populations had a very high superinfection incidence, with 6 populations showing a superinfection incidence of 100% in females: Taiwan, Thailand, Singapore, Philippines, Japan and Mauritius. The lowest incidence of superinfection (67%) was in females from Timor-Leste. There was substantial variation in the incidence of superinfections in males detected by qPCR. Excluding populations which had sample sizes less than 3, the PNG population had the highest superinfection incidence in males (100%) while Vanuatu had the lowest incidence (0%). When we analysed the populations with the largest sample sizes for each sex and considered only the *w*AlbB and superinfected individuals (N>5 per sex), a generalized linear model indicated a significant interaction among sex and population $X^2 = 1493.2$, df = 4, P < 0.001). In each population, the frequency of individuals infected with only *w*AlbB was lower in females than in males and sex differences were significant (P < 0.05) in four of the five populations by contingency tests. We also found 6 females that were singly infected with *w*AlbA, while rare uninfected (0% - 6.7%) mosquitoes were equally likely to occur in either sex.

Similar patterns to those presented in Table 3 were detected when the *w*AlbB and superinfection was identified through SNPs (Table 5). Although numbers are lower, these results also highlight the higher incidence of the superinfection in females compared to males in several populations.

## *Wolbachia* density comparisons

*Wolbachia* density was influenced by sex and the location where samples were collected (Fig 4). For *w*AlbA density, there was a significant effect of sex (F $_{(1, 378)}$ = 30.1413, P < 0.001) and population (F $_{(13, 378)}$ = 39.6288, P < 0.001) on density as well as a marginally significant interaction effect (F $_{(9, 378)}$ = 2.278, P = 0.008) when considering only superinfected individuals. For *w*AlbB, only the main effect of population was highly significant (F $_{(14, 493)}$ = 37.5341, P < 0.001) whereas the overall effect of sex (F $_{(1, 493)}$ = 3.0350, P = 0.082) was not significant and there was only a marginally significant interaction (F $_{(13, 493)}$ = 2.0186, P = 0.018). The PNG population had the highest density of *w*AlbA, while Sri Lanka had the lowest. Additionally, for *w*AlbB, the population of Singapore had the highest density, while Sri Lanka again had the lowest density.

*Wolbachia* density of the *w*AlbB infection did not differ in the singly infected and superinfected mosquitoes from Torres Strait (Fig 5A and 5B: F $_{(1, 132)}$ = 0.399, P = 0.529) and there was also no sex effect (F $_{(1, 132)}$ = 1.188, P 0.278) or interaction (F $_{(1, 132)}$ = 1.676, P = 0.198). For Vietnam females, there was also no difference between the infection type in density (Fig 5C; F $_{(1, 13)}$ = 1.048, P = 0.325).

**Table 2. Test set confusion matrix for sex prediction of mosquitoes based on ddRAD-seq markers identified from a Torres Strait dataset but tested against sexed samples from other populations.**

| | | Predicted sex | |
|---|---|---|---|
| | | Female | Male |
| Morphologically determined sex | Female | 89 | 3 |
| | Male | 0 | 32 |

**Table 3. *Wolbachia* infection status in *Ae. albopictus* females and males as assessed by qPCR screening.** Sexes were identified morphologically or through the ML sequencing-based approach as indicated.

| Population | Sex | Infection status (%) | | | | Number of individuals |
|---|---|---|---|---|---|---|
| | | *w*AlbA | *w*AlbB | *w*AlbA and *w*AlbB | Uninfected | |
| **Torres Strait** (multiple islands) | Male (M) | 0 | 52.4 | 47.6 | 0 | 63 |
| | Female (M) | 0 | 12.2 | 87.7 | 1.1 | 90 |
| **Malaysia** (Pahang, Johor, Selangor and Perlis) | Male (P) | 0 | 7.4 | 88.9 | 3.7 | 27 |
| | Female (P) | 0 | 1.1 | 97.8 | 1.1 | 89 |
| **Timor-Leste** | Male (M) | 0 | 28.6 | 71.4 | 0 | 14 |
| | Female (M) | 11.1 | 15.5 | 66.7 | 6.7 | 45 |
| **Indonesia** (Jakarta, Bandung and Bali) | Male (P) | 0 | 50 | 50 | 0 | 6 |
| | Female (P) | 0 | 0 | 100 | 0 | 36 |
| **Vietnam** (Ho Chi Minh City) | Male (M) | 0 | 50 | 50 | 0 | 2 |
| | Female (M) | 0 | 22.2 | 72.2 | 5.6 | 18 |
| **Taiwan** | Male (P) | 0 | 62.5 | 37.5 | 0 | 8 |
| | Female (P) | 0 | 0 | 100 | 0 | 10 |
| **Vanuatu** | Male (P) | 0 | 100 | 0 | 0 | 9 |
| | Female (P) | 0 | 14.3 | 85.7 | 0 | 7 |
| **Fiji** (Nadi) | Female (P) | 0 | 12.5 | 87.5 | 0 | 16 |
| **Thailand** (Chiang Mai, BKK) | Male (P) | 0 | 0 | 0 | 100 | 2 |
| | Female (P) | 0 | 0 | 100 | 0 | 7 |
| **Singapore** | Female (P) | 0 | 0 | 100 | 0 | 8 |
| **Philippines** (Manlia) | Male (P) | 0 | 0 | 100 | 0 | 1 |
| | Female (P) | 0 | 0 | 100 | 0 | 6 |
| **Japan** (Matsuyama) | Male (P) | 0 | 33.3 | 66.7 | 0 | 3 |
| | Female (P) | 0 | 0 | 100 | 0 | 3 |
| **Mauritius** | Male (P) | 0 | 50 | 50 | 0 | 2 |
| | Female (P) | 0 | 0 | 100 | 0 | 4 |
| **China** (Guangzhou) | Female (P) | 0.7 | 1.3 | 98 | 0 | 151 |
| **PNG** (Port Moresby and Madang) | Male (P) | 0 | 0 | 100 | 0 | 3 |
| | Female (P) | 0 | 0 | 97 | 3.0 | 33 |

Abbreviation: M, sexed based on morphology; P, sexed based on prediction using the ML model.

There was a significant association between *w*AlbA and *w*AlbB infections in superinfected *Ae. albopictus* females in most populations (Fig 6A–6F). Correlations between densities of the two infections were always positive and significant ($P < 0.05$) in the samples and regression analyses where the *w*AlbA density was used to predict the density of *w*AlbB were also always significant with positive slopes in each case (Fig 6). On the other hand, we found no associations between the density of the infections in males in the two populations with moderate sample sizes ($>10$) available for analyses (Fig 6G and 6H).

**Table 4. Test set confusion matrix of *Wolbachia* infection classification for *w*AlbB and superinfected (*w*AlbA and *w*AlbB) individuals from different populations, with values scored from the qPCR assay being predicted by the ML SNP-based method.**

| | | Predicted infection | |
|---|---|---|---|
| | | *w*AlbA and *w*AlbB | *w*AlbB |
| qPCR determined infection | *w*AlbA and *w*AlbB | 81 | 2 |
| | *w*AlbB | 5 | 7 |

**Table 5. Sex and strain distribution of *Wolbachia* infection status in *Ae. albopictus* using classification results.** Only samples with > 60% classification probability in both sex and *Wolbachia* strain prediction are included.

| Population | Predicted Sex | Number of individuals | | |
|---|---|---|---|---|
| | | Predicted *w*AlbB | Predicted *w*AlbA and *w*AlbB | Total |
| **China** (Guangzhou) | Female | 0 | 151 | 151 |
| | Male | 0 | 1 | 1 |
| **Indonesia** (Bandung) | Female | 0 | 12 | 12 |
| | Male | 0 | 2 | 2 |
| **Indonesia** (Bali) | Female | 0 | 8 | 8 |
| | Male | 0 | 2 | 2 |
| **Indonesia** (Jakarta) | Female | 0 | 17 | 17 |
| | Male | 0 | 0 | 0 |
| **Japan** (Matsuyama) | Female | 0 | 5 | 5 |
| | Male | 0 | 1 | 1 |
| **Malaysia** (Pahang, Johor, Selangor and Perlis) | Female | 0 | 61 | 61 |
| | Male | 0 | 15 | 15 |
| **Mauritius** | Female | 0 | 3 | 3 |
| | Male | 2 | 0 | 2 |
| **Fiji** (Nadi) | Female | 0 | 16 | 16 |
| | Male | 0 | 0 | 0 |
| **Philippines** | Female | 0 | 6 | 6 |
| | Male | 0 | 1 | 1 |
| **Singapore** | Female | 0 | 8 | 8 |
| | Male | 0 | 0 | 0 |
| **Sri Lanka** | Female | 0 | 0 | 0 |
| | Male | 0 | 1 | 1 |
| **Taiwan** | Female | 1 | 8 | 9 |
| | Male | 4 | 1 | 5 |
| **Torres Strait** | Female | 0 | 123 | 123 |
| | Male | 38 | 25 | 63 |
| **Vanuatu** | Female | 0 | 7 | 7 |
| | Male | 4 | 3 | 7 |
| **Vietnam** | Female | 0 | 6 | 6 |
| | Male | 1 | 0 | 1 |

## Discussion

### Population history

Overall, we found 23 mtDNA haplotypes in the Indo-Pacific samples of *Ae. albopictus* but no evidence of cryptic species. Minard et al. (2017) reported a novel cryptic species of *Ae. albopictus* in Vietnam which lacked *Wolbachia* [12]. Additionally, Guo et al. (2018) reported cryptic species of *Ae. albopictus* in China which were separated by substantial genetic distances from the other *Ae. albopictus* populations sampled from various regions within the country; *Wolbachia* was infrequent or absent in these populations of the putative cryptic species [11]. In contrast, we failed to detect the cryptic *Ae. albopictus* suggesting that they are not common across the Indo-Pacific region. This also implies that the cryptic lineage is not contributing to variation in the incidence of *w*AlbA and *w*AlbB across populations. Despite this, we found substantial variation in the incidence of *Wolbachia* infections, with a low incidence particularly in

 *Wolbachia* infections and mitochondrial DNA haplogroups in *Aedes albopictus*

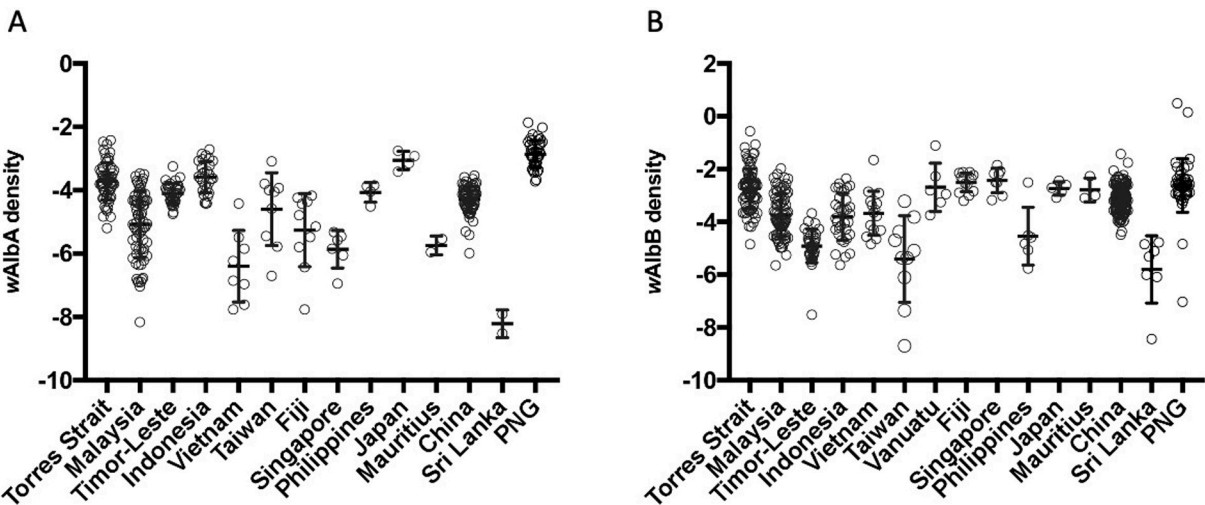

**Fig 4. *w*AlbA density (A) and *w*AlbB density (B) in *Aedes albopictus* from each population.** Variation in density across populations was analysed by a multi-way ANOVA (P < 0.001), focussing on female individuals carrying both infections. Vertical lines and error bars represent medians and 95% confidence intervals.

Timor-Leste despite the proximity of this sample to those we collected from Indonesia where both *w*AlbA and *w*AlbB infections were common.

Previous genetic analysis of *Ae. albopictus* populations from Asia with SNPs [15, 35] suggested three main regions of genetic similarity, one in East Asia, one in Southeast Asia (excluding Indonesia), and one in Indonesia. Some populations outside the native range showed clear signs of recent invasion from these regions; specifically, Mauritius from East Asia, and Fiji and Christmas Island from Southeast Asia [15]. A second study identified three major genetic groups: Torres Strait/Indonesia/Timor-Leste; East Asia/Southeast Asia/Fiji; and PNG/Vanuatu [35]. Gene flow from PNG into the Torres Strait was suggested by the higher co-ancestry between PNG and some Torres Strait genotypes.

Our mtDNA data confirmed three Asian range lineages of *Ae. albopictus*. The Indonesian lineage was highly differentiated from the others, indicating a common heritage between East

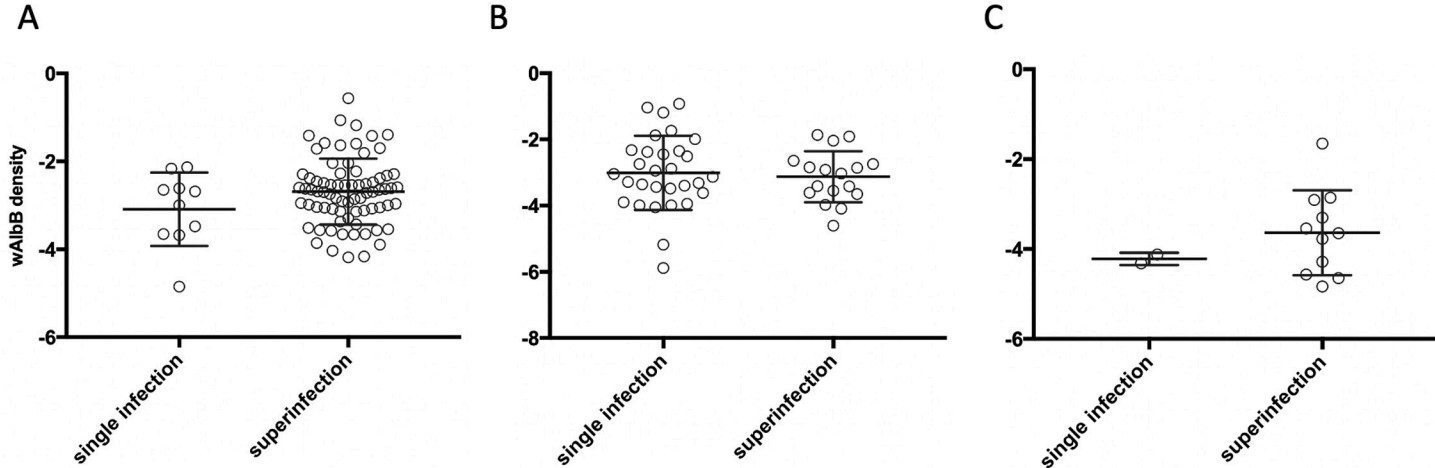

**Fig 5. Female (A) and male (B) *w*AlbB density in single-infected and superinfected *Ae. albopictus* from Torres Strait and female (C) *w*AlbB density in single- and superinfected *Ae. albopictus* from Vietnam.** Vertical lines and error bars represent medians and 95% confidence intervals.

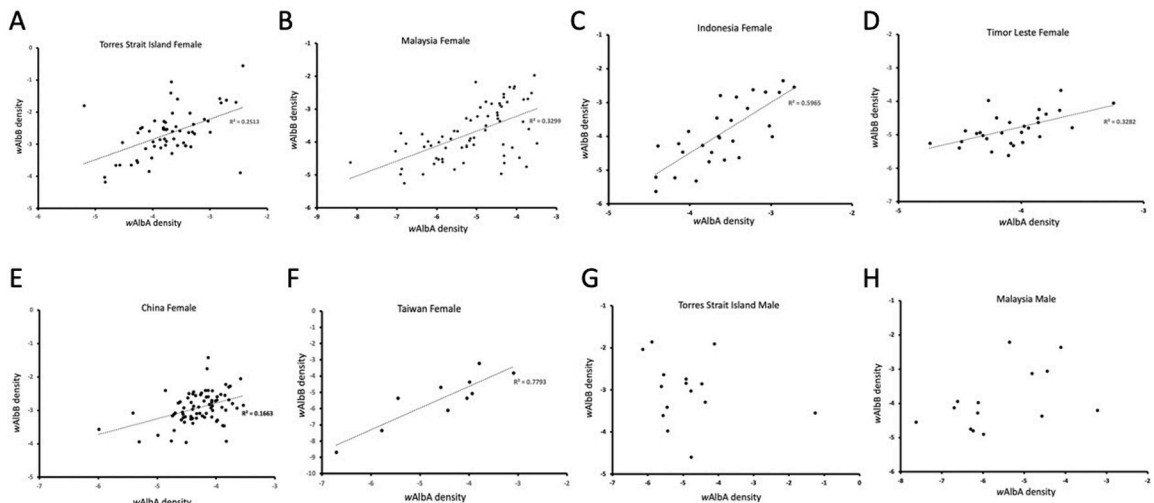

**Fig 6. *w*AlbA density and *w*AlbB density in individual females (A-F) and males (G, H) across populations.** Data are shown separately for samples from Torres Strait Island (A, G), Malaysia (B, H), Indonesia (C), Timor-Leste (D), China (E) and Taiwan (F). Regression lines are only included where significant (P<0.05) linear associations were detected.

Asian and Southeast Asian lineages. Shared haplotypes (H1) between PNG, Vanuatu, and Christmas Island provided the evidence of contemporaneous invasion of these populations. Shared haplotypes between Christmas Island and Sri Lanka (H9) as well as PNG, Fiji and other Southeast Asian populations (H11) grouped with all the other Southeast Asian haplotypes, suggesting recent co-ancestry between these invading populations and Southeast Asia. The presence of cryptic species in Sri Lankan populations had been previously hypothesized [15], but our phylogenetic analysis suggests there is no cryptic species within Sri Lanka populations. Nevertheless, the incongruence between the mtDNA and nuclear DNA results from the Sri Lanka samples remains of interest.

Initial genetic investigation of the Torres Strait showed that the invasion was most likely from Indonesia via a "stepping-stone" in the "Southern Fly" region of PNG [36,37]. Our results are accord with these findings. Haplotype 2 (H2) was found in all individuals from the Torres Strait and some individuals from the Philippines, Timor-Leste, and Indonesia, while this haplotype was not found in PNG. Considering the findings in Schmidt, Swan [35], where the Torres Strait nuclear genetic background was more like Indonesia than to Timor-Leste, Indonesia is supported as the most likely source of the Torres Strait invasion. H2 was connected to the East Asian haplogroup through a private haplotype from Timor-Leste (H8), which differed from H2 by one mutation step at nucleotide position 39. These results suggest an invasion scenario in which *Ae. albopictus* invaded Indonesia and Timor-Leste from East Asia, and subsequently colonised the Torres Strait.

## Variation in *Wolbachia* infections across populations

Although the *w*AlbA and *w*AlbB superinfection predominated in most populations, we did find some variation in the frequencies and densities of the two infections across populations and across sexes. Once at a high frequency, a superinfection should be maintained in a population unless transmission rates are low and host fitness costs are particularly high [19]. Invasion by a superinfection is expected to take place when the superinfection increases in frequency beyond a specific point dictated by host fitness costs, strength of CI and rate of maternal transmission. *Aedes albopictus* females with superinfections may have higher oviposition rates and

live longer [20]. However, the fitness effects of *Wolbachia* in *Ae. albopictus* appear to be complex and dependent on conditions [38] as they are in other insect systems where an infection that might appear to be deleterious can nevertheless still be beneficial in some situations [39]. Transmission of both the single infections and the superinfection appears to be close to 100% in *Ae. albopictus* [14] but transmission can also vary dramatically with environmental conditions as demonstrated by rearing *Aedes* under hot conditions [33].

Our findings reinforce the notion that there are sex differences in infection frequencies, and that the frequency of *w*AlbB as a single infection is higher in males than in females due to apparent loss of *w*AlbA in males [22]. Loss of an infection in the male sex is unlikely to alter the population dynamics of *Wolbachia* in mosquito populations that rely on transmission and selection through the female sex [40]. A single *w*AlbA infection can cause high levels of CI, allowing it to spread into an uninfected population [20,41], but ongoing selection may have led to a decrease of *w*AlbA in males, producing a decline of *w*AlbA-induced CI [20]. Incompatible matings effectively lower the fertility of infected males, leading selection to reduce infection density in males before sexual maturation, and wild-type superinfected *Ae. albopictus* males with very low *w*AlbA titres induce significantly lower levels of CI [6]. Superinfected females will remain compatible with males that lose the *w*AlbA strain as well as other males [20]. Alleles that favour loss of infection in males may therefore not necessarily be selected against in the absence of paternal transmission. In *Drosophila* there is also at least one CI infection that is often lost in males, although in this case the males retain a reduced capacity to cause CI in matings with uninfected females [42,43].

Apart from sex differences, we also found substantial variation in both density and incidence of the *w*AlbB and *w*AlbA infections across populations. We can compare our results to the *Wolbachia* infection status of *Ae. albopictus* in the Indo-Pacific region determined in previous studies [11,14,22,23,44–56] (S2 Table). These studies and ours highlight that the *w*AlbA/*w*AlbB superinfection or *w*AlbB single infection are present in most natural *Ae. albopictus* populations [22,23]. Single infections of *w*AlbA can occur in *Ae. albopictus* populations but appear variable in frequency at least in samples from Malaysia, China, and India [11,46,53]. Some caution is required in making comparisons between studies, given that the density of *w*AlbA can be low and its likelihood of detection may therefore be variable depending on the sensitivity of assays used by different laboratories. It is also possible that sample preservation following collection influences *Wolbachia* density and detection.

Several non-technical factors could contribute to the substantial variation in *Wolbachia* density across populations, including environmental conditions known to influence density such as temperature [57] and low levels of environmental antibiotics [58]. Age and life stage at collection also affect *Wolbachia* density [6,22,59], with age likely to be particularly important as well as contributing to any sex differences in density. In addition, *Wolbachia* density can be affected by nuclear factors as evident from the variation in density when introduced to different host backgrounds [60,61]. Variation in *Wolbachia* genomes could also contribute to density differences, given that between-population variation exists within *w*AlbB [62,63] and that closely related *Wolbachia* variants can differ in density [64].

*Wolbachia* infections can interact, or they can be independent in the host, but our density data provides no evidence of competition between the two *Wolbachia* strains. Although there was a lower density of *w*AlbA in mosquitoes as has previously been noted [6,41], we found no evidence that the presence or density of *w*AlbA had a negative effect on the density of *w*AlbB and vice versa. In superinfected mosquitoes, there was a consistent positive association of these infections in females from all samples and an absence of any pattern in males, highlighting the absence of antagonistic interactions among the infections.

## Strain characterization via SNPs

Our sexing data shows the usefulness of ddRAD-seq not only in population studies but also as markers that can be repeatedly used to investigate new issues as they arise. SNP markers have previously provided information on population structure and history across the Indo-Pacific region [15] and provided information on quarantine risk identification [65]. We also applied these markers to dissect patterns of selection on pesticide resistance markers [66] and have now used them to identify genomic features that have their sequencing depths associated with sex. With sufficient sequencing depth, RAD sequencing is a viable way to determine the sex of *Ae*. *albopictus*. Whereas sexing using morphology can be time consuming and impossible for immature samples and poorly preserved material, sex classification using ML can be automated as part of the processing workflow when working with RAD-seq data.

While ddRAD sequencing-based sexing worked well, we found a moderate amount of disagreement between the *Wolbachia* infection status from the SNP-based ML model and the status given from the qPCR assay. There are several possible explanations for this. First, the SNP features chosen for the ML model may not truly be able to separate the two *Wolbachia* strains. Second, cross-sample contamination of DNA may have led to the presence of *w*AlbA or *w*AlbB sequencing reads in non-infected samples. Lastly, extremely low density of *Wolbachia* may lead to screening inaccuracies when densities fall under the limit of detection by qPCR [67,68].

The results from the strain classification model were also heavily influenced by the relative proportions of classes in the training data. Due to being trained on a dataset where the number of superinfected individuals was approximately six times greater than the number of single *w*AlbB individuals, a sample with a low number of allelic observations, either due to low sequencing depth or low *Wolbachia* density, is more likely to be classified as superinfected than singly infected by *w*AlbA. Additionally, since there were insufficient singly infected *w*AlbA samples to train the model to classify singly infected *w*AlbA cases, new data with *w*AlbA would likely be classified as superinfected. Therefore, when classifying future samples from different populations, thus having different priors of *w*AlbA/*w*AlbB proportions, the existing model should perhaps be discarded, and the relative numbers of observed alleles from *w*AlbA/*w*AlbB could instead be used as a heuristic to classify samples.

## Applied implications

In the absence of cryptic species of *Ae*. *albopictus* and given the substantial variation in infection frequency, we suspect that *Wolbachia*-based interventions targeting this species are applicable across the Indo-Pacific region. Strains generated through transinfection and utilised in one location [69] may therefore be suitable for other regions as long as the nuclear background of the strain is altered to match that of the target population in case of local adaptation and local pesticide resistance levels [70]. However, given the preponderance of superinfected mosquitoes, it will be important to release individuals with new infections that are capable of invading local populations or supressing them via CI. We also emphasize the usefulness of SNPs in applied population studies more generally and as markers that can be repeatedly reanalysed to investigate new issues as these arise.

## Supporting information

**S1 Fig. PCA plot using the sequencing depths of significant regions showed two distinct clusters.**
(TIF)

**S2 Fig. Sex classification using a preliminary SVM model.**
(TIF)

**S1 Table. Estimates of Evolutionary Divergence between COI haplotypes.** The number of base substitutions per site from between haplotype sequences are shown. Standard error estimates are shown in the last column. Analyses were conducted using the Kimura 2-parameter model. The analysis involved 26 nucleotide sequences. All positions containing gaps and missing data were eliminated. There were a total of 623 positions in the final dataset. Evolutionary analyses were conducted in MEGA7.
(XLS)

**S2 Table. Field surveys of *Wolbachia* infections in populations of *Aedes albopictus* in the Indo-Pacific region from previous studies.**
(DOCX)

**S1 Data. Manuscript analytical dataset.**
(XLSX)

## Acknowledgments

We thank Tom Swan for providing the mosquito samples from Torres Strait and for morphological sex identification. We also thank Nancy Endersby and Venessa White for assistance with qPCR assays. This research was supported by use of the Nectar Research Cloud, a collaborative Australian research platform.

## Author Contributions

**Conceptualization:** Qiong Yang, Jessica Chung, Perran A. Ross, Ary A. Hoffmann.

**Data curation:** Qiong Yang, Jessica Chung, Thomas L. Schmidt, Jiaxin Liang.

**Formal analysis:** Qiong Yang, Jessica Chung, Ary A. Hoffmann.

**Funding acquisition:** Ary A. Hoffmann.

**Investigation:** Qiong Yang, Jessica Chung, Katie L. Robinson, Thomas L. Schmidt, Ary A. Hoffmann.

**Methodology:** Qiong Yang, Jessica Chung, Katie L. Robinson.

**Project administration:** Qiong Yang.

**Resources:** Qiong Yang, Jessica Chung, Thomas L. Schmidt, Ary A. Hoffmann.

**Software:** Qiong Yang, Jessica Chung, Jiaxin Liang.

**Supervision:** Ary A. Hoffmann.

**Validation:** Qiong Yang, Jessica Chung.

**Visualization:** Qiong Yang.

**Writing – original draft:** Qiong Yang, Jessica Chung, Ary A. Hoffmann.

**Writing – review & editing:** Katie L. Robinson, Thomas L. Schmidt, Perran A. Ross, Jiaxin Liang.

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
