## [Decision Letter · Decision Letter 0]

9 Feb 2022

Dear Dr. Yang,

Thank you very much for submitting your revised manuscript "Sex-specific distribution and classification of Wolbachia infections and mitochondrial DNA haplogroups in Aedes albopictus from the Indo-Pacific" for consideration at PLOS Neglected Tropical Diseases. Your manuscript was reviewed by the editorial board and by the independent reviewers. Based on the reviews, we are likely to accept this paper for publication, providing that you modify the manuscript according to the additional recommendations made the reviewers. 

Sincerely,

Pattamaporn Kittayapong, Ph.D.

Associate Editor

Louis Lambrechts, Ph.D.

Deputy Editor

Reviewer's Responses to Questions

**Key Review Criteria Required for Acceptance?**

**Methods**

-Are the objectives of the study clearly articulated with a clear testable hypothesis stated?

-Is the study design appropriate to address the stated objectives?

-Is the population clearly described and appropriate for the hypothesis being tested?

-Is the sample size sufficient to ensure adequate power to address the hypothesis being tested?

-Were correct statistical analysis used to support conclusions?

-Are there concerns about ethical or regulatory requirements being met?

Reviewer #1: 1. Mitochondrial COI as a DNA barcode and The trimmed 623 bp sequence was analysed with Geneious 9.18 software (Kearse et al., 2012) to investigate SNP variation among samples. These methods may not be sufficient to identify species.

2. ML sex classification with ddRAD-seq ，How reliable is this method?

3. Wolbachia detection via qPCR assay，What are the advantages and confirmations of this method compared with general PCR.

Reviewer #2: (No Response)

**Results**

-Does the analysis presented match the analysis plan?

-Are the results clearly and completely presented?

-Are the figures (Tables, Images) of sufficient quality for clarity?

Reviewer #1: Figure 6. wAlbA density and wAlbB density in individual females (A-F) and males (G, H) across populations.Why do some have linear correlation and some don't? What is the significance of this difference? Why?

 Test set confusion matrix for sex prediction of mosquitoes based on SNP markers identified from a Torres Strait dataset but tested against sexed samples from other populations. How accurate is the prediction accuracy of SNP markers?

Reviewer #2: (No Response)

**Conclusions**

-Are the conclusions supported by the data presented?

-Are the limitations of analysis clearly described?

-Do the authors discuss how these data can be helpful to advance our understanding of the topic under study?

-Is public health relevance addressed?

Reviewer #1: modification

Reviewer #2: (No Response)

**Editorial and Data Presentation Modifications?**

Reviewer #1: (No Response)

Reviewer #2: (No Response)

**Summary and General Comments**

Reviewer #1: Sex-specific distribution and classification of Wolbachia infections and mitochondrial DNA haplogroups in Aedes albopictus from the Indo-Pacific， it is Scientifically significant.

Reviewer #2: Yang et al analysed population genomic data and tested for cryptic species in 160 Ae. albopictus sampled from 16 locations across Indo-Pacific region and found that Ae. albopictus can partitioned into three groups, but with no evidence of cryptic species. They also show wAlbA and wAlbB infections showed markedly different patterns in the sexes, with most population carrying superinfection and males having more single wAlbB infection than females. These results provide important background information for developing Wolbachia to control dengue transmitted by this mosquito vector. The manuscript is well written. Below are some minor comments. 

Figure 4, What is the host gene used to normalize the Wolbachia copy? This information should be included in method. There is no PNG (the highest one) density in Fig.4A. Statistic information should be provided in the figure. 

Why not validate the sex through assay of Nix gene by PCR? Is this more straightforward and easier than ddRAD-seq? 

The age of sample can be a big factor affecting the variation of Wolbachia infection between sexes as male can lose wAlbA with aging. This probably is more important reason than others in term of higher rate of wAlbB in males than females in the tested samples.

PLOS authors have the option to publish the peer review history of their article (what does this mean?). If published, this will include your full peer review and any attached files.

Reviewer #1: No

Reviewer #2: No

Figure Files:

Data Requirements:

Reproducibility:

References

---

## [Editor Report · Decision Letter 1]

14 Mar 2022

Dear Dr. Yang,

We are pleased to inform you that your manuscript 'Sex-specific distribution and classification of Wolbachia infections and mitochondrial DNA haplogroups in Aedes albopictus from the Indo-Pacific' has been provisionally accepted for publication in PLOS Neglected Tropical Diseases.

Best regards,

Pattamaporn Kittayapong, Ph.D.

Associate Editor

Louis Lambrechts, Ph.D.

Deputy Editor

---

## [Editor Report · Acceptance letter]

12 Apr 2022

Dear Dr. Yang,

We are delighted to inform you that your manuscript, "Sex-specific distribution and classification of Wolbachia infections and mitochondrial DNA haplogroups in Aedes albopictus from the Indo-Pacific," has been formally accepted for publication in PLOS Neglected Tropical Diseases.

Best regards,

Shaden Kamhawi

co-Editor-in-Chief

Paul Brindley

co-Editor-in-Chief
